# ZERO-SHOT REWARD SPECIFICATION VIA GROUNDED NATURAL LANGUAGE

## ABSTRACT

Rewards in reinforcement learning are difficult to design for most tasks and often require access to underlying state information. The alternatives to reward signals are usually demonstrations or goal images which can be labor-intensive to collect. Describing the goal as a text description is a low effort way of communicating the desired task to an agent. However, goal-text conditioned policies so far have been trained in scenarios either with rewards that assume access to states or with labeled expert demonstrations. In this work, we devise a model that leverages CLIP to ground objects in a scene described by the goal-text paired with spatial relationship rules to provide an off-the-shelf reward signal using only raw pixels as input. We distill the policies learned with this reward signal on several tasks to obtain a single text conditioned policy that can be applied to solve new tasks at deployment. We show the effectiveness of this framework on a set of robotic manipulation tasks.

## 1 INTRODUCTION

Previous efforts have explored image-based goal specification, with significant successes in visual navigation and manipulation tasks (Pathak et al., 2018; Nair et al., 2018; Fu et al., 2018; Singh et al., 2019). Yet existing image-based goal specification paradigms are limited because they are typically limited to a particular scene *instance* in an environment, whereas a *semantic* goal comprises multiple possible scene configurations. Reinforcement learning offers one of the most appealing premises in the study of AI: from a reward signal alone, algorithms which learn optimal policies that maximize expected reward can learn to perform navigation, dexterous manipulation, and host of other impactful tasks. However, discovering or specifying a reward function for a given task is often a very challenging problem, especially when one is considering agents that can learn from uninstrumented environments, e.g., from raw image observations alone. We wish to have an agent that can learn purely from pixels, with no access to the underlying state of the environment at any point during learning or task execution. Achieving this goal without access to an instrumented reward function has been exceedingly challenging.

One can use image-based reward specification to cause a robot agent to navigate to a particular chair next to a specific tall plant, but that agent may not always succeed at the generic task of "go to a chair next to a tall flowering plant": e.g., if the goal specification image shows a red chair next to

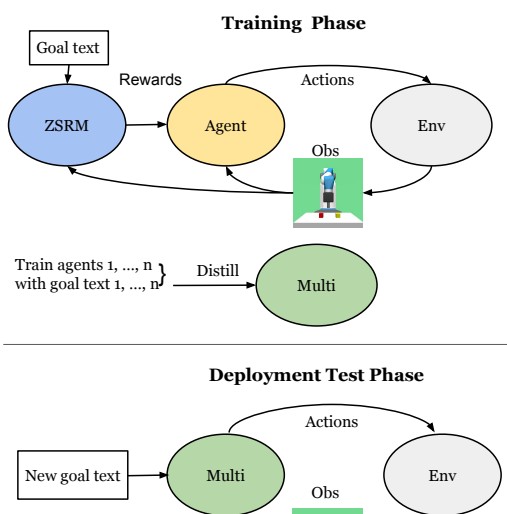

Figure 1: We devise a method that learns tasks using only goal text from user and images from the environment at training time. We train several agents on several tasks and distill their policies into a multi-task policy that is conditioned on goal text and current observation. The distilled policy is able to generalize to new unseen tasks at test time. We assume no access to environment reward, state, demonstrations, or goal images.

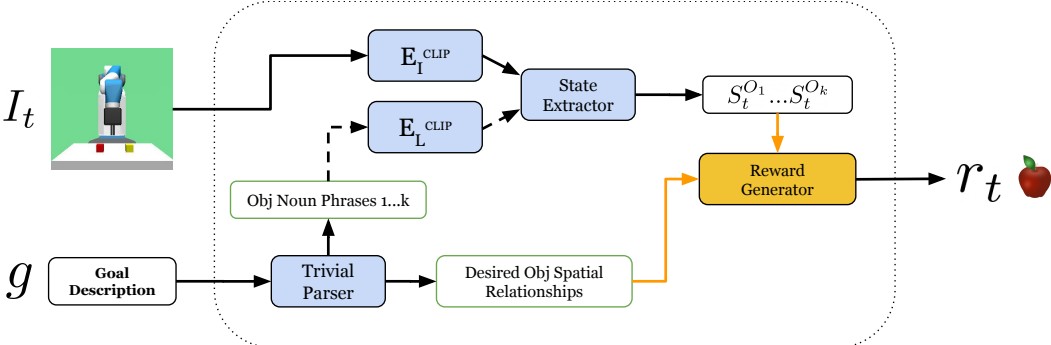

Figure 2: We compute reward of an image observation conditioned on goal language description, with no access to state, demonstrations, or goal images. We first parse the goal description into object noun phrases and desired object spatial relationships. We then leverage CLIP's image and language encoders $E_I$ and $E_L$ to locate the object noun phrases in the image observation. Using the object states and the desired object spatial relationships, reward is generated to train the agent for the desired task in its environment.

a plant with a yellow flower the agent may navigate away from a scene with a blue chair next to a red flower, depending on the model's underlying image representation. To be sure that the true goal is properly specified irrespective of the invariances of the model's underlying perceptual representation, a user may have to provide a set of goal image examples that cover the variation of the target concept, potentially a very expensive undertaking to collect or generate.

We advocate *semantic reward specification via grounded natural language*, where a user describes a goal configuration in the world using a natural language description referring to entities in the world. This direction has long been a "holy grail" of AI research and a presumed capability of AI science fiction—the ability to instruct a robot with natural language—yet attempts have been limited by the state of the art in grounded language perception. It also falls under the general umbrella of leveraging large-scale passive data to bootstrap embodied learning, where we rely on language as a means to provide necessary reward shaping missing in visual data alone.

Several previous efforts train reward functions or policies that take natural language as input for goal description (Oh et al., 2017; Bahdanau et al., 2018; Zhou & Small, 2020; Goyal et al., 2020; Fu et al., 2019; Hermann et al., 2017; Shao et al., 2020). They all however rely on reward signals that have access to state of the system or demonstrations of the task distribution they are training on. There are works that use human videos to learn reward functions to train their agent with (Shao et al., 2020; Sermanet et al., 2018; 2016), but they require a curated dataset of humans performing the tasks.

Recently however, the advent of large-scale multimodal training data together with large capacity language and vision deep learning models has significantly advanced the state of the art. A steady series of innovations have advanced grounded language modeling, from early work on multimodal translation and fusion models (Barnard et al., 2003; Quattoni et al., 2007; Guadarrama et al., 2016), to large-scale joint embedding models (Frome et al., 2013; Radford et al., 2021), to the plethora of multimodal transformer models currently under investigation (Su et al., 2019; Lu et al., 2019; Chen et al., 2019; Hu & Singh, 2021). CLIP, in particular, demonstrated a transformative advance on zero-shot object recognition (Radford et al., 2021).

We could specify a goal state to a robot by simply offering a description of the goal configuration in natural language and using the CLIP embedding dot product with an observed image to evaluate proximity to goal state. Surprisingly, this can work in simple cases, for examples see the top example in Figure 5. But for more complex goals performance can be poor, especially with complex captioned scenes involving spatial relationships and on domains that differ from web images (as shown in (Radford et al., 2021)) and the bottom example in that figure.

To overcome these limitations, we propose a factored grounding of 'what' vs 'where/how' aspects of goal state (Figure 2), and offer a novel spatial-salience scheme for the former. We argue that existing (e.g., CLIP-like) models can be used to ground the *what* aspects of a goal quite effectively, including appropriate attribute and concept-level generalization, while a separate *where/how* module can ground spatial relationship aspects of goal configuration. After training many tasks with this

reward scheme we learn a language conditioned policy that can execute new tasks in a zero-shot fashion – from a natural language description of task – without having to train a new policy for every new task.

In summary, our contributions are as follows:

1. A zero-shot reward function to learn a set of robotic manipulation tasks on raw pixels without access to state. (Zero-shot in the sense that no goal images or human demonstrations are provided at training time or test time.)

2. A distilled language-conditioned multi-task policy on several tasks learned using the zero-shot reward function which is able to deploy new unseen tasks.

## 2 METHOD

### 2.1 BASE ZERO-SHOT REWARD MODEL: DOT-PRODUCT VISUOLINGUISTIC REWARD

Our base model is the most intuitive way to use the CLIP model to compute reward. Our base model simply computes reward by taking a dot product between the goal language input feature and image observation feature through CLIP's language and image encoders respectively: $r_t = E_I(I_t) \cdot E_L(g)$ A subset of tasks can be learned with this reward model. We visualize the limits of this model on two environments in Figure 5.

### 2.2 FULL ZERO-SHOT REWARD MODEL: SPATIALLY GROUNDED VISUOLINGUISTIC REWARD

**CLIP-Saliency for Phrase Grounding**   We have developed a simple method which factors phrase grounding and spatial relationship processing, which can be used to specify a reward function that operates only on image content using a natural language description. Figure 2 illustrates our full model leverages CLIP's encoders in a very different way than the base model. Our model first parses the goal language description $g$ into object noun phrases $g_{o_1}...g_{o_k}$ and the desired object interaction. The object noun phrases are passed through CLIP's language encoder and the current image observation is passed through CLIP's image encoder. We use the language encodings as class embeddings of each object noun phrase to perform a saliency analysis (using Grad-CAM (Selvaraju et al., 2017)) on the image encoding of the observation on the last convolutional layer (specifically, the ReLU layer of CLIP's ResNet-50 backbone). Saliency models such as Grad-CAM generally output a heatmap of the features that indicate a class exists in the current image input. The state extractor in Figure 2 computes the "state" of each object using the argmax of the saliency heatmap. See Figure 3.

**Spatial Grounding Heuristics**   Our model determines task completion using a spatial grounding rule based reward function that takes the object pixel coordinates of one or more camera views and desired spatial relationship as input. Our spatial heuristics can be used to describe a broad set of manipulation tasks to push or place objects to semantic locations. A reward of 1 is given to the RL algorithm for every timestep that the conditions of the reward criteria are met for a given spatial relationship. The oracle reward uses true x,y,z spatial positions of the objects to determine if the desired spatial relationship

Table 1: Spatial Grounding Heuristics

| Spatial Relationship | Reward Criteria |
|---|---|
| Obj1 on the left of Obj2 | $O_x^2 > O_x^1$ |
| Obj1 on the right of Obj2 | $O_x^1 > O_x^2$ |
| Obj1 on top of Obj2 | $|O_x^1 - O_x^2| < \epsilon_1 \ \& \ O_y^2 < O_y^1 < O_y^2 + \epsilon_2$ |
| Obj1 below Obj2 | $|O_x^1 - O_x^2| < \epsilon_1 \ \& \ O_y^1 < O_y^2 < O_y^1 + \epsilon_2$ |
| Obj1 in between Obj2, Obj3 | $min(O_x^2, O_x^3) < O_x^1 < max(O_x^2, O_x^3)$ |
| Obj1 in front of Obj2 | $O_{x2}^1 > O_{x2}^2$ |
| Obj1 behind Obj2 | $O_{x2}^2 > O_{x2}^1$ |
| Obj1 close to Obj2 | $\|O_{xy}^1 - O_{xy}^2\|_2 < \epsilon$ |
| Obj1 inside of Obj2 | $\|O_{xy}^1 - O_{xy}^2\|_2 < \epsilon$ |

is reached and also outputs 1 to the RL algorithm for every timestep the conditions of the task are met.

Our spatial heuristics are fully defined in Table 1. The first set (left of, right of, on top of, below, and in between) assume coordinates in a front camera view and the semantics are defined in that camera view with positive y in the upward direction and positive x in the right direction. The second set (in front of & behind) has access to a left camera view with coordinate x2 pointing towards the right

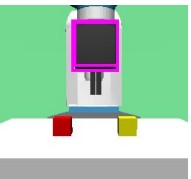 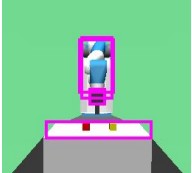 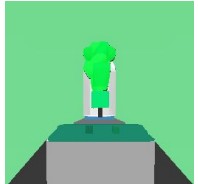 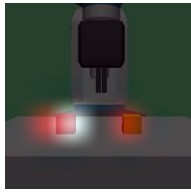 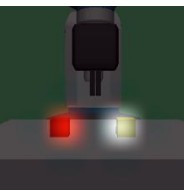

Figure 3: a) Mask R-CNN object detection results b) Mask R-CNN detection results on far view c) Mask R-CNN instance segmentation results on far view d) Grad-CAM result with 'red block' text input (white being highest intensity) e) Grad-CAM result with 'yellow block' text input (white being highest intensity)

which is towards the front in the first camera view and y2 pointing upward similar to the first camera view. The second camera is needed to know if the object is placed in the front or behind another object correctly. The third set (close to & inside of) require access to a front camera view with a 45 degree downward tilt towards the ground. This camera view is needed to see if the object is getting closer to another object in two orthogonal directions at once where as a left only or front view only allows you to determine one dimension of closeness. For "inside of" a 45 degree camera helps the agent see if the object is going inside another object without occlusion. The $\epsilon$ threshold for "inside of" is much smaller than for "close to" since the centroid can come much closer when an object goes inside a container object.

## 2.3 Language Conditioned Multi-task Policy

Now that we have a method for learning tasks in a zero-shot reward fashion, we would like to be able to not require training a new policy for every new task and ideally have a language conditioned multi-task policy that takes in the goal text description of an unseen task and executes the task without needing any new samples from the environment.

To approach this goal, we first learn several tasks via reinforcement learning using our zero-shot reward model. We then create a large dataset of the rollouts of those polices and pair each trajectory with the goal text description of the tasks it was trained to learn. We then use behavioral cloning (supervised learning for predicting actions) to learn a policy that takes images and text goal task description as input and actions as target outputs. We use CLIP's language encoder as the text goal embedding that is fed to the multi-task policy. In addition we use image augmentation techniques to aid behavioral cloning in learning more robust policies.

## 3 Experiments

### 3.1 Phrase grounding and Base Zero-shot Reward Model Visualization

Object detectors are one way to extract object states, however, they are usually not used off the shelf and need to be fine-tuned with in domain data. In Figure 3 we show pretrained Mask R-CNN (He et al., 2017) outputs on different camera views for the block stack environment. As you can see in the first two subfigures, the blocks are not proposed as objects with Mask R-CNN from both far and close camera views. This is a demonstration of the need for in-domain fine-tuning of object detectors to work in the environment you want to use. Our Grad-CAM output from CLIP however, highlights exactly the objects we are interested in from the object noun phrases that describes each object in the last two subfigures.

The CLIP model has been trained on a much larger dataset than Mask R-CNN and has a language component that allows us to request for the location of the object noun phrase of interest. With Mask R-CNN however, even if the object proposals were good, it would not necessarily classify objects it has not been trained for and therefore

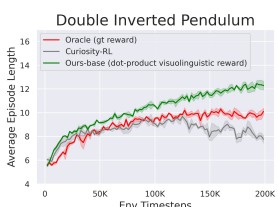

Figure 4: We showcase the Base Reward Model in the Double Inverted Pendulum doing slightly better than oracle reward. We average results over 3 seeds per task.

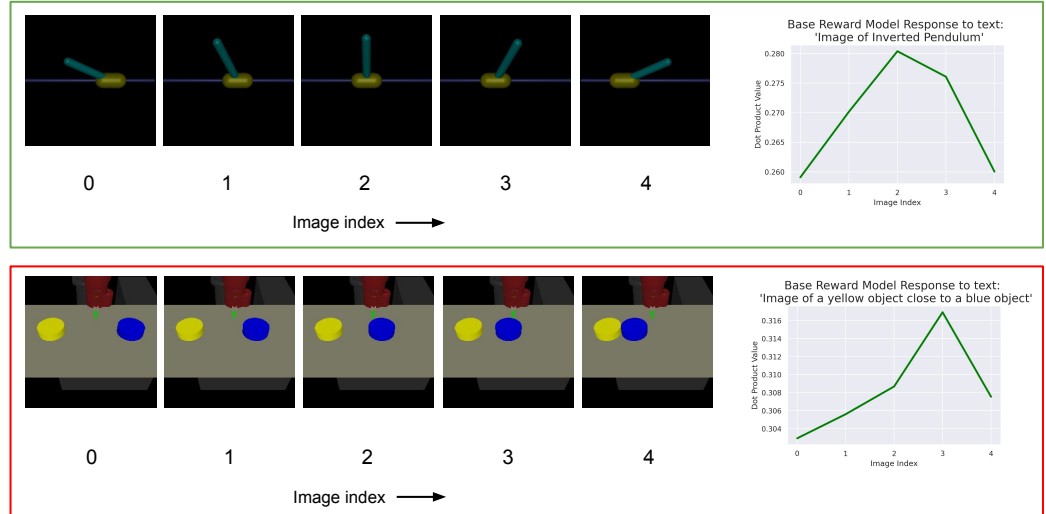

Figure 5: We visualize the results of the base reward model which is a trivial dot product between the goal language description and image observation. The top row (green box) displays a successful utilization of the base reward model and the bottom row (red box) shows a failure case. The x-axis represents image index.

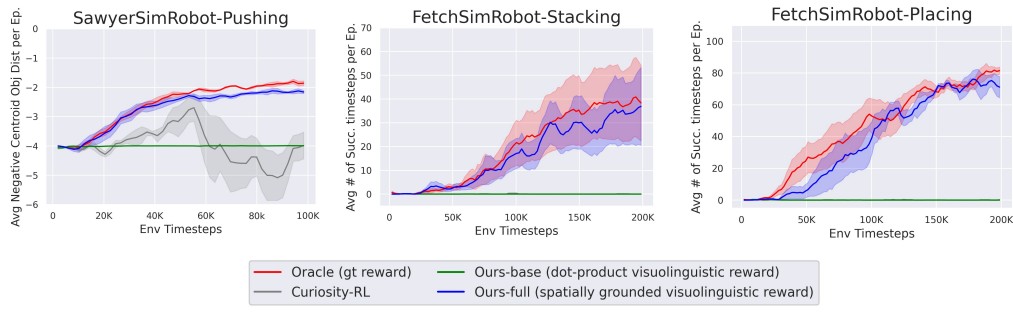

Figure 6: We showcase our Full Reward Model performing almost as well as oracle reward on SawyerSimRobot-Pushing (pushing a blue puck close to a yellow puck), FetchSimRobot-Stacking (stacking a yellow block on top of a red block), and FetchSimRobot-Placing (placing a yellow block on the right of a red block). We average results over 3 seeds per task.

not output a filtered set of object proposals. In other words, we would not know which object is which if the detector hasn't been trained with the label of objects we care about.

In Figure 5 we show what our base reward model outputs on two goal descriptions: 1. inverted pendulum 2. yellow object close to a blue object. For the first goal description we observe that the dot product increases as the pendulum becomes more inverted from either side as desired. For the second goal description we observe that as the blue object gets closer to the yellow object the dot product increases except for the closest image where it dips which results in an undesired output. We observed through these two goal descriptions and many others that the base reward model is not sufficient for recognizing object spatial relationships. The encoders are good at identifying what objects are in the image however, which is what we leverage for our full reward model.

## 3.2 FULL ZERO-SHOT REWARD MODEL RESULTS

In Figure 6 we show how our full reward model performs on three manipulation tasks. We train each task using our full zero-shot reward model output as reward for the PPO reinforcement learning algorithm (Schulman et al., 2017). We then train with the oracle reward that has access to state and the base zero-shot reward model for comparison. Curiosity reward (Pathak et al., 2017; Burda

et al., 2018) is also used as our baseline since it only has access to images similar to our method for computing reward, but has only been successful for videogames such as Atari and Mario or Locomotion where exploring new states leads to progressing through the task (going further in levels of game for example). It is less privileged however, in that it does not use language input for task specification.

Curiosity learns the pushing pucks close together task but then starts learning separation of the pucks which reemphasizes that curiosity is only useful for tasks where exploring new dynamics leads to going farther in the task. Curiosity also has some trouble learning the double inverted pendulum task because the dynamics of the pendulum swinging can be hard to predict and therefore have misleading higher reward. Curiosity also does not learn the other manipulation tasks (stacking and placing) as those are more complex tasks that are harder to reach by exploration. For Double inverted pendulum (Fig. 4) our base reward model does better than oracle by chance which we speculate is because the oracle reward was originally designed for state input and was not tuned for learning image to reward mapping. Our base reward model fails for pushing, stacking, and placing, which take "an image of a yellow block on top of a red block", "an image of a yellow object close to a blue object", and "an image of a yellow block on the right of a red block" as language input for those tasks respectively. For pushing, stacking, and placing our full reward model performs almost as well as oracle sparse reward which has privileged access to state.

The oracle reward function for Double-Inverted-Pendulum is alive bonus minus distance penalty minus velocity penalty. The oracle for SawyerSimRobot-Pushing is a sparse reward that outputs one when the centroid distance between two pucks are below a threshold. The oracle reward for FetchSimRobot-Stacking is a sparse reward that outputs one when a yellow block is within a horizontal and vertical threshold distance of a red block. The oracle reward for FetchSimRobot-Placing is a sparse reward that outputs one when a yellow block is correctly placed on the right of a red block. See Figs. 3 & 5 for image observation examples of FetchSimRobot and SawyerSimRobot.

## 3.3 MULTI-TASK POLICY RESULTS

In order to avoid having to train a new policy for every new task we want to learn, we train a multi-task policy with a set of training tasks and then show that it can generalize to a set of unseen test tasks by leveraging CLIP's language model to encode the goal text description of the tasks as conditioning input to our multi-task policy.

In the FetchSimRobot environment we train 18 tasks with PPO each for 200K environ-

|  | Seen distribution | | | | Unseen distribution | | | |
|---|---|---|---|---|---|---|---|---|
|  | train tasks | | test tasks | | train tasks | | test tasks | |
| (episode reward stats) | mean | s.e. | mean | s.e. | mean | s.e. | mean | s.e. |
| No Conditioning | 22.71 | 1.13 | 12.74 | 0.89 | 17.60 | 1.08 | 9.01 | 0.87 |
| Primitive Code Cond. | 35.06 | 1.29 | 16.60 | 0.99 | 25.03 | 1.14 | 12.66 | 0.85 |
| Language Cond. | 43.62 | 1.30 | 20.08 | 1.08 | 31.28 | 1.23 | 14.79 | 1.02 |

Table 2: Multi-task Policy Performance Results: We report multi-task average and standard error performance across all 18 training tasks and all 18 test tasks (unseen object color in goal text or image) for seen and unseen initial state distributions with no conditioning, primitive code conditioning, and language conditioning. The values reported are the episode reward averaged across 50 seeds per task policy rollout. The tasks are robotic arm manipulation of objects to different target semantic relationships of each other in the FetchSimRobot environment from only images. While the policies were trained using our zero-shot reward function, the reward metric reported is oracle reward.

ment steps with our zero-shot reward model described in the previous section and show generalization results for 18 unseen test tasks by training a language conditioned policy with behavior cloning on rollouts of the 18 training tasks. We collect 5000 timesteps per task which is around 50 trajectories. We do this for pickplace-left, pickplace-right, and stack tasks for different object combinations. The object training colors are red, green, and yellow, and the object test color is blue. So the multi-task policy has never seen blue block in text input or image input.

For each task we average episode rewards over 50 seeds. We then average across all training tasks to get the train metric and across all test tasks to get the test metric respectively. We compare our language conditioned policy with a policy trained without task labels, and one trained with a primitive

code as the conditioning input. The latter baseline simply labels the training tasks with integers 0 to 17. Since it only has primitive codes for the training tasks it uses CLIP's language embedding to find the closest primitive code it was trained with to the test language goal description.

We use the same oracle reward that has direct access to state in figure 6 to measure the performance of our multi-task models. Every timestep the objects are in the correct target text description the agent is rewarded 1 point. The objects never start in the correct target state. Therefore it always takes several timesteps for the policy to move the objects to the correct state. All episodes timeout after 100 timesteps or if the one of the objects falls off the table. In RL training and in policy rollout collection for behavior cloning each object has a one block width of variation in starting point location. We tested on same distribution of start point variation interval (seen initial state distribution) and double the start point variation interval (unseen initial state distribution).

In Table 2 we see that no conditioning has the lowest performance as expected since there is full ambiguity in what task to perform given only an image. However, since the no conditioning model has been trained across many different object colors going to different target states it has learned general displacement of blocks to different semantic locations randomly that can sometimes be moved to the correct semantic location by chance during testing. For the training tasks it has higher performance than testing tasks since it can memorize to execute some of the training tasks that have almost the exact same initial states sampled during testing as those in the behavior cloning dataset and thereby execute correctly as memorized.

The primitive code conditioned model has much higher performance than the no conditioning model since it can disambiguate what task it needs to execute. Surprisingly language conditioning performs significantly better than primitive code conditioning in training task performance. We conjecture this is due to language conditioning being a more structured representation of the task, providing a better loss manifold from which the model can learn a better perceptual motor representation of the task distribution. In addition the training tasks are sampling initial states from a continuous distribution meaning almost no two policy rollouts have the same initial state. So some amount of generalization needs to be had even in the seen initial state distribution to have high performance on the training tasks.

The language conditioned model is significantly better in test task performance over the primitive code conditioning since it can use the language embedding to infer the target task determined by the goal text description. The primitive code conditioning model has access to the language embedding only to determine the closest training primitive code to the target task description. It compares the text description embedding of all the training tasks to the test task text description embedding and chooses the training task primitive code corresponding to the smallest L2 distance. It's interesting to observe that the primitive code conditioning does better than no conditioning on the test task because the closest primitive code can sometimes lead to a successful execution of the test task since it has some signal towards the correct task. One such example is that the test goal description task of "a blue block on the right of a yellow block" is mapped to the primitive code of the training goal description task of "a red block on the right of a yellow block".

In Table 2 we also show the same trend in the models performing at double the interval of initial starting locations of the objects seen at training (unseen intial state distribution): the no conditioning policy performing the poorest on train and test tasks, the primitive code conditioned model performing significantly better on both, the language conditioned model performing the best at training and test tasks via its language structured specification of tasks.

We train the multitask policy with ResNet18 image encoding that is concatenated with CLIP language encoding both of which are pushed through an fc layer before concatenation. The concatenated vector is then inserted into two fc layers before predicting actions. We apply an L2 regression loss for continuous action space. The policy is trained using Adam optimizer with AMS grad with learning rate of 1e-4. The images are augmented with PyTorch RandomResizedCrop of 0.95 to 1.0 area and 0.98 to 1.02 aspect ratio randomization and resized to original image dimensions of 128x128. All the multitask models are trained for 300 epochs.

# 4 RELATED WORK:

**Goal conditioned policies** Goal conditioned policies allow the user to specify the agent's goal. States (Schaul et al., 2015; Andrychowicz et al., 2017) and Images (Pathak et al., 2018; Nair et al., 2018; Fu et al., 2018; Singh et al., 2019) are one way of specifying goal. However, they assume that user has access to a photo or state of final task given to the agent. We assume no access to goal images or state to minimize human labor and instrumentation, and use language which provides a natural form of supervision.

**Goal text conditioned policies** Several previous efforts train reward functions or policies that take natural language as input for goal description (Oh et al., 2017; Bahdanau et al., 2018; Zhou & Small, 2020; Goyal et al., 2020; Fu et al., 2019; Hermann et al., 2017; Shao et al., 2020). They all however rely on reward signals that have access to state of the system or demonstrations of the task distribution they are training on.

**Learning reward functions** There are works that use human videos to learn reward functions to train their agent with (Sermanet et al., 2018; 2016; Shao et al., 2020). We however, don't need a curated dataset of humans performing the tasks we want our agent to train with. Having humans perform all tasks we are interested in may not scale well with the amount of labor needed for recording and curating those datasets. CLIP has the advantage of learning a caption model from 400 million image, text pairs from the publicly available sources on the internet WIT.

**No Environment Reward** There have been recent work on methods that use no reward signal from the environment to train for specific tasks. Two notable ones are Curiosity (Pathak et al., 2017) and DIAYN (Eysenbach et al., 2018). Curiosity has only been successful on video games with discrete action space and DIYAN has mostly been successful on direct state input. Our method however, can learn a subset of manipulation tasks in continuous action space on raw pixels and has the advantage of specification from user goal text.

**Utility of large vision language models in RL** There has been recent work that leveraged CLIP to do many complex manipulation tasks (Shridhar et al., 2021). They however, have access to labeled expert demonstrations for training their policy. We assume no access to demonstrations or goal images at training or test time.

# 5 DISCUSSION

In this work, we presented a method for learning a set of object manipulation tasks without access to state of the system to compute reward from text goal description alone. Our method doesn't use goal images or demonstrations at training time or test time. We devised a zero-shot reward model that leverages a language vision model (CLIP) that has been trained on a very large dataset of captioned images on the internet to compute progress (reward) towards a goal text description. We use this zeroshot-reward model to collect data on many tasks to then supervise a language conditioned multi-task policy that can execute new tasks without need of extra training.

Our work opens avenues for many directions such as learning spatial relationships from data or leveraging separate datasets for spatial relationships. Pioneering work along these lines was proposed in (Guadarrama et al., 2013; Golland et al., 2010), who showed that spatial relationships could be grounded in a learning-based system. We encourage pursing the integration of these approaches to the grounding of spatial prepositions with the noun-phrase grounding capability we have demonstrated based on our CLIP-salience method.Our current method also does not specify state of the system other than semantic position such as object pose or closed door.

Finally we must address the ethical perils inherent in our leverage of models trained on large-scale vision and language datasets. Such datasets are well known to suffer from dataset bias that can cause failure or unintended harm (Buolamwini & Gebru, 2018). While the near-term risks appear to be limited with the robotic applications presently envisioned, practitioners should continuously monitor systems for bias against underrepresented groups and ensure that robotic systems work across all socioeconomic domains. Techniques for bias assessment and debiasing should be employed whenever possible to ensure this remains the case.

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
