# OpenReview forum: "Zero-Shot Reward Specification via Grounded Natural Language"
_ICLR.cc/2022/Conference — ICLR 2022 Submitted_

### Official Review · Reviewer_GnSJ · 2021-11-02

**Correctness:** 3
**Technical Novelty And Significance:** 3
**Empirical Novelty And Significance:** 2
**Recommendation:** 5
**Confidence:** 4

**Main Review:**

*Strengths*

The paper tackles and interesting and important problem in 0-shot reward specification. Solving 0-shot reward specification, particularly from language is essential to getting RL to work in new environments. Additionally the approach proposed in this work is creative and performs well in the domains considered from images.

*Weaknesses*

The weaknesses in this work are (1) the generality of the proposed approach, (2) the experimental domains/comparisons, and (3) coverage of prior work.

(1) The paper proposes an interesting approach to 0 shot reward specification in explicitly mapping an instruction to nouns and spatial relations, using CLIP to localize a noun, then using the position and spatial relation with heuristics to define the reward. However I think there are a number of limitations of this approach that the paper should discuss further.

First, parsing the instruction into objects and into a fixed set of spatial relations does seems limited to programmatic language and the specific tasks in the paper (e.g. in the paper the instruction “an image of a yellow block on top of a red block" directly maps to the "on top of" rule). But this likely will not be trivial in the case where the instruction is in the form of natural language. Even for the "inverted pendulum" example shown earlier in the paper, its not clear how a parser would map that to objects and a specific spatial relation.

Second, the approach of extracting the object positions with Grad-CAM is a clever idea, but it does seem very specific to environment, tasks, and viewpoints considered in the paper. In a more interesting visual scene with more than 2 objects its not clear that the Grad-CAM would provide as good object localization. The paper also mentions that for this environment very specific camera viewpoints are needed for the GRAD-CAM localization to work. Moreover there are many tasks (e.g. opening a cabinet) where its not clear how to heuristically define a reward based on the Grad-CAM localization, or what part of the cabinet the saliency map will map to.

It does seem that most of the reward specification is actually coming in the form of hardcoded rules, and the main use of CLIP is in localizing objects.

(2) The experiments show that on some simple table-top manipulation tasks the proposed approach can nearly match an Oracle reward function. This is an interesting result, however says more about the ability to use CLIP to localize objects than it does about the particular reward function approach, since the heuristic reward is the same between both this method and the Oracle. The curiosity driven baselines also are not that informative since they do not get the language instruction. Some more interesting comparisons would be (a) how the method compares to goal-image specifications and (b) how it might compare to using an off the shelf image-captioning approach. It would also be valuable to see if this approach can handle (a) more complex natural language instructions, and (b) more interesting environments with more complex visuals than the 2 solid colored blocks.

(3) There are a number of missing related works [1,2,3,4] which also learn language conditioned skills on robots. Particularly [1-3] also learn to ground language to goals or rewards.

[1] MacGlashan et al. Grounding English Commands to Reward Functions. RSS 2015.
[1] Arumugam et al. Grounding natural language instructions to semantic goal representations for abstraction and generalization. Autonomous Robots 2019.
[3] Nair et al. Learning Language-Conditioned Robot Behavior from Offline Data and Crowd-Sourced Annotation. CoRL 2021
[4] Lynch and Sermanet. Language conditioned imitation learning over unstructured data. RSS 2021.

**Summary Of The Paper:**

The paper presents a hybrid learning/rule-based approach to language-conditioned reward specification, particularly for robotic control tasks from images. The core contribution of the work is in specifying the reward 0-shot. Specifically, the method parses a language instruction into an object noun phrase and desired spatial configuration. Then it uses CLIP embedding of each object phrase as a query, and uses Grad-CAM to get a saliency map in the image which is used to localize the target object. Finally a set of heuristics are used to compute the reward for a particular spatial configuration given the target object(s) coordinates. The reward can then be used to train either single or multi-task policies. Experiments suggest that in Simulated Fetch robot domains this approach can nearly match an Oracle reward.

**Summary Of The Review:**

Overall the paper presents an interesting and creative solution to the important problem 0-shot language specification for robot manipulation from images. However, the proposed method seems to be task/environment/viewpoint specific, and the paper should further discuss these limitations. The paper can also improve its experimental domains, baseline comparisons, and coverage of related work.

---

> ### Author Response · Authors · 2021-11-23
> **Authors Response to Reviewer GnSJ**
>
> We thank the reviewer for the constructive feedback and glad they also find our defined problem statement “essential” and found our work “creative” and “performing well in domains considered from images”. We address their concerns in detail below.
>
> >*“How the method compares to goal image specifications”*
> - On reviewer’s suggestion, we have just ran FetchRobotSim results on a version of VICE [Fu et.al. 2018] as a privileged baseline that has access to true spatial relationship of models and show the results in this link: https://drive.google.com/file/d/1j6uOOvyy-cQybN7PG1e7ARnEBseXekt2/view?usp=sharing
> - We train VICE [Fu et.al. 2018] reward model as a goal classifier that has access to true goal images of the task such as “red block on top of yellow block” and use images that aren’t in the correct goal configuration as negative images. We show that this privileged baseline performs as well as oracle reward and our method for comparison.
>
> >*“parsing the instruction into objects and into a fixed set of spatial relations does seems limited to programmatic language and the specific tasks in the paper”*
> - We agree that a rule based parser is limiting and we did not focus on it as we thought it wasn’t the challenging part of the problem statement as there are methods out there to parse out the rules automatically.
> - However, it is possible to make a parser-free version of the spatial visuolinguistic reward method by doing a CLIP-style training of an end-to-end reward model. We leave it for future work.
>
> >*”Even for the "inverted pendulum" example shown earlier in the paper, its not clear how a parser would map that to objects and a specific spatial relation.”*
> - If the goal task involves object spatial relationships we use the spatial visuolinguistic model, otherwise we use the dot product visuolinguistic model.
>
> >*“In a more interesting visual scene with more than 2 objects it’s not clear that Grad-CAM would provide as good object localization”*
> - Our method is not limited to 2 objects as we have experimented with more. One example is Obj1 in between Obj2, Obj3 described in table 1.
>
> >*”Moreover there are many tasks (e.g. opening a cabinet) where it’s not clear how to heuristically define a reward based on the Grad-CAM localization.”*
> - The spatial visoulinguistic reward is useful mostly for spatial manipulation tasks. We focused on spatial tasks however, as that seemed to be the most limiting factor for using CLIP to compute rewards in a zero-shot fashion. We explicitly specify this in the paper, and will clarify it further in the introduction.
> - However, that being said, we have preliminary results for tasks like opening and closing doors using the dot product visuolinguistic reward but we leave it for future work.
>
> >*“Missing related work”*
> - Thank you, we will add those related works to our paper. None are in the same zero-shot problem statement as us, but they are great additions to our related work thanks.

---

### Official Review · Reviewer_PiKW · 2021-11-02

**Correctness:** 3
**Technical Novelty And Significance:** 2
**Empirical Novelty And Significance:** 2
**Recommendation:** 3
**Confidence:** 4

**Main Review:**

The idea to use natural language grounded in the scene as the reward function is an interesting idea and a goal for language grounded RL. However, I have some concerns/questions.

1. Definition of zero-shot. The reward function is determined by the heuristics defined in table 1. These heuristics encode the knowledge about the goal. So, the reward function is not entirely zero-shot as the heuristics encode the structure of the goal.

2. The use of natural language is limited. Originally, I expected to see the reward function that is directly computed from the CLIP image embed and text embed. It is reasonable that the paper shows the dot product doesn’t represent the spatial relationship well. However, the final proposal, i.e. using a CLIP-based saliency map and a separate spatial module, doesn’t leverage the vision-language model well. The use of CLIP here is closer to an object detector. While this paper showed that the pretrained mask RCNN doesn’t provide good detection, this still doesn’t show change the fact that CLIP is used as an object detector. In this case, any color detector may work as well too. Then, what does a pretrained vision-language model help in the reward specification? This separation also limits the possibility to extend to other language descriptions beyond spatial relationships.

3. CLIP-base result in Figure 6. Even though the reward using a simple dot product may not be good in all cases, why does Figure 6 show the base result to be always flat? It doesn’t seem likely that RL agent that uses this reward function doesn’t learn anything from the beginning.

4. Alternative CLIP baseline? While the dot product version doesn’t work. I’m wondering if there is an alternative formulation. For example, turns the reward specification into classification of objects of different spatial relationships. This would remove the need for separate spatial heuristics.

**Summary Of The Paper:**

This paper demonstrates how to use pretrained CLIP model as a reward function for an RL agent. This approach enables flexible goal specification using language. The proposed reward generation method uses CLIP to identify the relevant objects and a separate module to compute the reward based on a specified spatial relationship. This separation produces a better reward function than using the dot product between the embedded image vector and language vector.

**Summary Of The Review:**

The idea of the paper is interesting. However, the final implementation doesn’t really need the pretrained vision-language model.

---

> ### Author Response · Authors · 2021-11-23
> **Authors Response to Reviewer PiKW**
>
> We thank the reviewer for the constructive feedback and address the concerns in detail below. We numbered the responses in reference to your numbered concerns / questions.
>
> > *“1. ...the reward function is not entirely zero-shot as the heuristics encode the structure of the goal...*
> - We will make it clear in the paper that we mean zero-shot in the sense that zero demonstrations, zero goal images, and zero ground truth data is used at training and testing.
>
> > *“2. ... the final proposal, i.e. using a CLIP-based saliency map and a separate spatial module, doesn’t leverage the vision-language model well...This separation also limits the possibility to extend to other language descriptions beyond spatial relationships...”*
> - We argue that depending on the task one can use the spatially grounded visuolinguistic or dot product visuolinguistic reward depending on the task. For spatial tasks, we recommend using the spatially grounded visuolinguistic reward.
>
> > *“3. ...Even though the reward using a simple dot product may not be good in all cases, why does Figure 6 show the base result to be always flat?...”*
> - The dot product visolinguistic result isn’t entirely flat. The score is too low to see that it does learn a tiny bit on some of the spatial tasks.
>
> > *“4. Alternative CLIP baseline? While the dot product version doesn’t work. I’m wondering if there is an alternative formulation. For example, turns the reward specification into classification of objects of different spatial relationships. This would remove the need for separate spatial heuristics.”*
> - We have just ran FetchRobotSim results on a version of VICE [Fu et.al. 2018] as a privileged baseline that has access to true spatial relationship of models and show the results in this link: https://drive.google.com/file/d/1j6uOOvyy-cQybN7PG1e7ARnEBseXekt2/view?usp=sharing
> - We train VICE [Fu et.al. 2018] reward model as a goal classifier that has access to true goal images of the task such as “red block on top of yellow block” and use images that aren’t in the correct goal configuration as negative images. We show that this privileged baseline performs as well as oracle reward and our method for comparison.

---

### Official Review · Reviewer_fEgi · 2021-11-02

**Correctness:** 3
**Technical Novelty And Significance:** 2
**Empirical Novelty And Significance:** 2
**Recommendation:** 3
**Confidence:** 4

**Main Review:**

The paper tackles the problem of specifying goal configurations via language text. This is a more general (and therefore, more challenging) setting that is agnostic to the specific instance of the target objects/configurations presented via the goal image. The empirical evidence presented by the authors seems to support their claims.

However, given below are some of my concerns with the paper.
+ The goal text descriptions seem to be severely under-specifying tasks. For instance, consider an example of a goal text from the paper: “image of an inverted pendulum”. This doesn’t really say a lot about the desired task and it might hold true for several configurations within a delta tolerance of the actual goal state. That makes me wonder if the kind of goal-text descriptions being considered in the paper can be treated as good proxies for specifying tasks? I would imagine this problem becoming severe with task complexity when the approach is deployed to settings with slightly more complicated state spaces and task configurations. For instance, consider grid-world navigation. What would a goal description that leads to a high correlation between grounding of objects in the current observation and progress towards the goal be for the task of going to location X in the grid?

+ The Introductory section of the paper makes the case against reward engineering by saying that it utilizes the underlying state information. The spatial grounding heuristics used in the proposed approach are getting computed based on the GT locations of objects in the scene. Doesn’t that count as having access to state information and “engineering” rewards? How do the authors reconcile this? Moreover, computing the rewards based on whether spatial relationships (as decoded from the goal text) are being satisfied or not involves the use of multiple camera views: this brings in additional complexity from a real-world deployment perspective.

+ It is not clear how their approach fundamentally differs from the reward structure used here (https://arxiv.org/pdf/1904.04404.pdf). The reward is derived from the IoU and classification accuracies of predictions of the model and the policy gets trained depending on how well the agent’s perception can localize the target object. One might say that they warm start their training with expert trajectories which the authors do not, but to be fair, the cited paper works with a much more complex state space (3D indoor scenes). Ignoring that, the general principle of deriving a reward based on localization/grounding of a target semantic concept holds true here as well.

+ The paper is missing a lot of critical information such as:
  + There is no information of how the goal text descriptions are derived. Is it by using some sort of captioning model on the image goal description or are they manually annotated for every task?
  + The paper doesn’t have any details on how the grounding of semantics and spatial relationships from the goal text onto the visual observations translates to numeric rewards. That seems to be an important step in their approach.

Also listed below are some minor suggestions and clarification questions:
+ In Sec 2.2, the authors say that the goal text is parsed into object noun phrases and object interactions. What does the latter mean? Are these the spatial relationships that the referred objects must satisfy?
+ Did the authors try fine-tuning the Mask-RCNN? Is the comparison shown in Fig. 3 between non fine-tuned Mask-RCNN and CLIP? If so, do the authors have a sense for the degree of overlap between the vocabulary of classes between MaskRCNN/CLIP and their environments?
+ In Sec 3.2, the language makes it seem like your model is progressively trained using GT reward first and then your proposed visio-linguistic rewards. This could get a little confusing, consider rewording to ensure that it’s clear that one of them is your approach and the others are baselines.

**Summary Of The Paper:**

The paper proposes a framework to train policies for tasks that are specified via language text without the use of any expert trajectories or underlying state information to engineer reward functions. The authors leverage a state-of-the-art visual grounding model (CLIP) to ground object nouns from the text into the current visual observation in order to derive a proxy for the rewards signal. Through some toy experiments they demonstrate performance of a simple baseline that implements this concept as well as enhancements to the baseline (saliency grounding + spatial relationship based rewards) that overcomes drawbacks of the naive approach. They also train their language-conditioned on several tasks to obtain a generic policy that can learn to solve an arbitrary task, given paired text descriptions and trajectory rollouts from the training tasks.

**Summary Of The Review:**

Overall, I am not entirely convinced that the approach of simply grounding text descriptions onto image observations can serve as a reliable proxy for rewards. In addition to that, the paper, in its current format, has a lot of missing details. Therefore, my vote is to reject this paper

---

> ### Author Response · Authors · 2021-11-23
> **Authors Response to Reviewer fEgi**
>
> We thank the reviewer for the constructive feedback and address the concerns in detail below.
>
> > *“The goal text descriptions seem to be severely under-specifying tasks [...] “image of an inverted pendulum”.”*
> - We would like to highlight that there is a tradeoff between how easy it is for the user to specify a goal vs. how precisely they can define the goal. For instance, a goal image or video or demonstration can specify a task very precisely but is very difficult for the user to specify as it essentially requires the user to have performed the task already. For instance, in case of an inverted pendulum, the user would have to figure out how to invert the pendulum and then take a photo to communicate it to the agent. Majority of the work in the community is focused on specifying tasks via goal image or demonstration as we discuss in the prior work sections. In this paper, we study how to make it easier for the user to specify tasks. We agree with the reviewer that in general text descriptions intrinsically can be underspecified for a large subset of tasks --- however, since we work with natural language, a user can always give more details in the task description.
> - For the particular example of “inverted pendulum”, the task is to make sure the pendulum is inverted. So if the agent inverts the pendulum within some threshold of perfect inverted angle, a delta angle, we still count it as success.
>
> >*“What would a goal description that leads to a high correlation between grounding of objects in the current observation and progress towards the goal be for the task of going to location X in the grid?”*
> - For such navigational tasks, one could change the goal text to what would describe the goal location as we would communicate to a human: go to “the green lamp next to the couch”. As the agent explores its environment, when it gets closer to the green lamp next to the couch, our dot product visuo-linguistic reward will give higher reward. This is because CLIP is very good at identifying when objects are in an image.
>
> >*“The spatial grounding heuristics used in the proposed approach are getting computed based on GT locations of objects in the scene. Doesn’t that count as having access to state…”*
> - We do not assume any access to ground truth locations of objects in the scene. They are estimated with CLIP saliency and are sometimes inaccurate. This is in contrast to most existing methods in manipulation which compute rewards by either engineering / instrumenting the environment or training an object detector for the specified task with in-domain data.
>
> >*“How goal text descriptions are derived”*
> - Our captions are manually specified as the most simple description we thought of with no engineering to find the best caption for the task.
>
> >*“How the grounding of semantics and spatial relationships from the goal text onto the visual observations translates to numeric rewards”*
> - Our reward is simply one when the desired spatial relationship is satisfied and zero otherwise.
>
> >*“In Sec 2.2 … what is meant by object interactions. Are these the spatial relationships?”*
> - Yes, these are the spatial relationships.
>
> >*“Did the authors try fine-tuning Mask-RCNN… ”*
> - No, we did not fine-tune Mask-RCNN as it would violate our problem statement. We do not assume any access to object ground truth locations, detection boxes or segmentations to fine tune Mask-RCNN with. Figure 3 was meant to show that object detectors currently cannot be used reliably to detect objects off the shelf and there is no guarantee of vocabulary overlap with the task a user would want. CLIP has the advantage of open vocabulary capacity.
>
> >*“In Sec 3.2 the language makes it seem like your model is progressively trained using GT reward first and then your proposed visio-linguistic rewards”*
> - Thanks for the recommendation; we will change the writing to clarify we never have access to ground truth reward during any point of training or testing.
>
> >*“It is not clear how their approach fundamentally differs from the reward structure used here (https://arxiv.org/pdf/1904.04404.pdf)...the general principle of deriving a reward based on localization/grounding of a target semantic concept holds true here as well.”*
> - We will cite this work, thank you! However, the setup of this [Yang et.al.] paper is very different from ours. They are not focused on solving a language specified task but instead on the problem of actively moving in a 3D environment for visual recognition of a target object (embodied vision recognition).
> - Our main focus is deriving task-specific rewards in a zero-shot fashion at test time just from a language instruction. To recap, the concrete differences are: (a) They are focused on moving around to improve object recognition, while we solve manipulation tasks. (b) They do not use language instructions at test time. (c) User does not specify a task to be solved in their case.

---

### Official Review · Reviewer_b31Y · 2021-11-03

**Correctness:** 3
**Technical Novelty And Significance:** 3
**Empirical Novelty And Significance:** 2
**Recommendation:** 6
**Confidence:** 2

**Main Review:**

The work proposes a simple and novel solution to the zero-shot reward specification problem which is an important step towards getting RL agents to work in new environments and tasks. The experiments show promising results and support the author’s claims.

Weakness:
- Experiments have been performed on only simple scenarios and environments.
- The natural language queries are simple and the authors don't talk about translating to more complex tasks
- Using Clip based text embedding with a dot product in the reward function performs worse than the heuristic-based reward function, but since the embedding based approach is translatable, more experiments could have been performed to understand why the reason for the lower performance
- Missing details on the construction of goal-text. Is it constructed in a way that is easy for the parser to be parsed?
- Missing related works in the space of learning language conditioned reward function or learning to perform language conditioned tasks [1].

[1] Fu J, Korattikara A, Levine S, Guadarrama S. From language to goals: Inverse reinforcement learning for vision-based instruction following. arXiv preprint arXiv:1902.07742. 2019 Feb 20.

**Summary Of The Paper:**

The paper proposes an approach for zero-shot reward specification using natural language groundings without expert demos and state information. The work proposes using CLIP’s image+language encoder to find the saliency maps and separately encoding the target locations for the object. These objects are identified by parsing the task sentence. The work proposes automatically getting a reward function by again using the sentence. The work claims that automatically parsing the goal text and using a heuristic to get a reward function works better than using the CLIP's language encode to goal text and get the rewards with a dot product.

**Summary Of The Review:**

The experiments in the paper are on toy examples and the approach proposed has several limitations for it to translate to other tasks, for example, the complexity of the environment and the task specification. But the techniques proposed are simple and can be extended to more complex cases with further research.

---

> ### Author Response · Authors · 2021-11-23
> **Authors Response to Reviewer b31Y**
>
> We thank the reviewer for the constructive feedback and glad they found our approach interesting and promising for stimulating new lines of research.
>
> >*”Missing details on the construction of goal-text”*
> - Our goal-texts are manually specified as the most simple description we thought of with no engineering to find the best goal-text for the task.
>
> > *Missing related work [1]*
> - We actually did cite this paper. Please refer to pages 2 and 8.
>
> > *The experiments are on toy examples*
> - We agree with the reviewer that it would be nice to show results in more complex environments and tasks. However, note that even the current environments are significantly difficult because we do not assume any access to rewards, demonstrations, or goal images and manipulate objects directly from image space. We hope that the first results in this challenging problem statement will be scalable to more complex tasks.

---

### Decision · Program_Chairs · 2022-01-20

**Decision:**

Reject

**Comment:**

This manuscript describes a method that turns sentences into reward functions by recognizing objects, parsing sentences into a simple formalism, and then grounding the parse in the recognized objects to form a reward for an agent.

1. The title and much of the manuscript are written in a way that reviewers found confusing. It would seem from the title and most of the text that the method integrates language models, CLIP specifically, into RL in a novel way to provide zero-shot rewards. But this is not the case. CLIP is used purely as an object detector. Yes, the method requires a good object detector and CLIP provides that, but any good object detector that can handle arbitrary phrases would have done.

2. The overall setup of the work: extract the state of the world and then parse sentences to formulate rewards by grounding parts of the parse into parts of the world state has been explored widely in robotics. Reviewers provided citations going back several years, but many others exist.

I would encourage the authors to rewrite the manuscript around their central contributions and downgrade their use of CLIP and language models in general to a minor technical footnote. Similarly refocusing related work on the robotics literature and demonstrating how this approach differs and improves on the state of the art there could result in a strong contribution.